# Revisiting the Necessity of Graph Learning and Common Graph Benchmarks

## Abstract

Graph machine learning has enjoyed a meteoric rise in popularity since the introduction of deep learning in graph contexts. This is no surprise due to the ubiquity of graph data in large scale industrial settings. Tacitly assumed in all graph learning tasks is the separation of the graph structure and node features: node features strictly encode individual data while the graph structure consists only of pairwise interactions. The driving belief is that node features are (by themselves) insufficient for these tasks, so benchmark performance accurately reflects improvements in graph learning. In our paper, we challenge this orthodoxy by showing that, surprisingly, node features are oftentimes more-than-sufficient for many common graph benchmarks, breaking this critical assumption. When comparing against a well-tuned feature-only MLP baseline on seven of the most commonly used graph learning datasets, one gains little benefit from using graph structure on five datasets. We posit that these datasets do not benefit considerably from graph learning because the features themselves already contain enough graph information to obviate or substantially reduce the need for the graph. To illustrate this point, we perform a feature study on these datasets and show how the features are responsible for closing the gap between MLP and graph-method performance. Further, in service of introducing better empirical measures of progress for graph neural networks, we present a challenging parametric family of principled synthetic datasets that necessitate graph information for nontrivial performance. Lastly, we section out a subset of real-world datasets that are not trivially solved by an MLP and hence serve as reasonable benchmarks for graph neural networks.

## 1 Introduction

In recent years, graph machine learning has become a pillar of the deep learning ecosystem (Kipf & Welling, 2017; Velickovic et al., 2018; Hamilton et al., 2017; Xu et al., 2018). This is largely unsurprising, as a plethora of data types (e.g. social networks, conceptual relationships in knowledge, natural science phenomena, visual interactions) naturally exhibit graph structures that go beyond the typical translational or sequential connections found in images or language. Graph neural networks (GNNs), and the myriad of associated variants, provide a natural interface for working with and interacting with such data through the graph neighborhood aggregation layer (typically implemented via models of message passing (Xu et al., 2018)), which enables node features to interact with each other in a principled manner.

The key justification for these methods is that taking the underlying graph structure into consideration should improve performance on downstream tasks. Otherwise, one could just use a standard graph-agnostic neural network operating on the features alone to achieve equally proficient results. Taken at face value, it is difficult to argue against such a reasonable justification: for common intragraph prediction tasks such as link prediction or node classification, this condition seems like an obvious necessity for a well-designed model. This, in turn, justifies many improvements to the core graph aggregation layer. While there have been many developments upon the original GNN architecture (Velickovic et al., 2018; Hamilton et al., 2017; Liu et al., 2020; Wu et al., 2019a), we highlight that many of these approaches adopt graph-centric theories as justification (e.g. equivariant expressivity through the WL test (Bodnar et al., 2021), geometric relaxations of discrete structures (Ganea et al., 2018), or topological rewiring of bottlenecks (Di Giovanni et al., 2023)),

further underscoring the importance of graph structure as critically valuable for graph machine learning methods.

In our paper, we call this assumption into question by demonstrating that, for several popular benchmarks, this graph structure justification does not hold. In particular, we find that for these widely used datasets a thoroughly tuned multi-layer perceptron (MLP) operating solely on node features can sufficiently close the gap with graph-based methods. Our results dispute the efficacy of many existing benchmarks as well as the assumption of the necessity of graph learning. As such, our work takes a crucial step towards better quantifying progress in the field of graph machine learning. Concretely:

1. We test seven commonly used datasets and identify five for which a well-tuned feature-only MLP drastically improves upon previously presented baseline performance (reducing error by up to 70%). In five of said datasets, the tuned MLP beats at least one graph neural network from the literature and, in one case, the tuned MLP beats all but one of the graph-based methods.

2. We show that these five datasets have features that naturally encode the graph structure and hence only marginally benefit from use of the graph. We do so by performing a careful analysis of how said features impact performance and comparing the effects for datasets that benefit from graph learning with the effects on those that do not.

3. Finally, we propose several alternatives to accurately benchmark the effect of graph aggregation layers. In particular, we start from first principles by designing a class of synthetic Watts-Strogatz (Watts & Strogatz, 1998) graph dataset tasks that, by construction, rely either solely or far more on graph structure than node features. Additionally, we section out a subset of real-world benchmarks that are not solved by an MLP, indicating that they require graph structure for nontrivial improvement.

## 2 Related Work

Our work addresses graph learning at large, both its necessity (or lack thereof) for various datasets, as well as the need for stronger benchmarking in the development of graph neural network methods. As such, we highlight related work in graph deep learning as well as work that discusses problems in the field.

**Classical graph neural networks** Modern graph neural network literature began with the paper Kipf & Welling (2017), which introduced Graph Convolutional Networks (GCNs). The ubiquity of graphs and the power of large scale machine learning quickly transformed graph machine learning into an incredibly large and diverse field (Wu et al., 2019b). That being said, these methods were not without their problems. Wu et al. (2019a) pointed out that many of the modifications being made to graph neural networks (GNNs) were superficial and presented a way to considerably simplify GNNs. Additionally, Huang et al. (2020) showed that a very simple graph neural network construction with few parameters could match most state-of-the-art models that had an order of magnitude more parameters. Such papers proved useful at forcing the field to step back and think about what methodological changes were truly useful. In a similar sense, our paper aims to push for more rigorous research and careful proposals by noting that seemingly fundamental assumptions such as usefulness of the graph must be questioned and further investigated to isolate good benchmarks and a proper context for graph machine learning methods.

**Graph benchmarks** A large number of important graph benchmarks focus on intragraph prediction tasks. These primarily feature link prediction (when a model must predict whether two nodes are connected or not) and node classification (when a model must predict one of a discrete set of classes for a given graph node) (Kipf & Welling, 2017). Out of the myriad datasets that exist, a particular collection of seven has been used widely throughout a lot of the classic graph learning literature. These are comprised of four co-purchase/co-authorship datasets: Amazon Computers (Shchur et al., 2018), Amazon Photo (Shchur et al., 2018), Coauthor CS (Shchur et al., 2018), and Coauthors Physics (Shchur et al., 2018), and three citation network datasets: Cora (Sen et al., 2008), Citeseer (Sen et al., 2008), Pubmed (Sen et al., 2008).[1] These network datasets are canonical because they are thought to be good examples of real-world graphs and social

---

[1]These datasets are provided under a CC0 1.0 Universal license.

networks. In particular, they exhibit the small world phenomenon (Kleinberg, 2000) while also being of a fairly manageable size. Due to their widespread presence in the graph learning literature as benchmarks, we focus on analyzing these datasets in this paper.

**Topological methods for graph neural networks** Recent work has shown that the underlying graph structure can commonly exhibit defects that significantly degrade downstream graph learning performance. Conceptually, the underlying graph structure can be "adjusted" (in terms of topological or geometric properties) to help improve downstream performance. Concretely, Oono & Suzuki (2019); Alon & Yahav (2021); Rusch et al. (2023) noted that the message passing algorithms used in graph neural nets suffered from a variety of ailments, including *under-reaching*, *over-squashing*, and *over-smoothing*; i.e., that topological *bottlenecks* in the graph impeded the flow of information. Topological fixes to these issues prominently featured *graph rewiring* methods, using graph curvature (Nguyen et al., 2023), topology (Sonthalia et al., 2023; Di Giovanni et al., 2023), and spectral analysis (Gasteiger et al., 2019; Karhadkar et al., 2023; Black et al., 2023; Arnaiz-Rodriguez et al., 2022). These methods attempt to de-couple the initial graph structure from the computational graph structure by adding or deleting edges.

**Geometric graph neural networks** Another way of improving graph learning is through geometric graph neural networks, in which the graph (and accompanying node features) are embedded in a metric space with "richer" geometry than straightforward Euclidean space. For example, Bronstein et al. (2017) brought attention to the fact that several kinds of complex data benefit from manifold considerations. In particular, Nickel & Kiela (2017) has shown that graphs with a hierarchical structure benefit from representation through hyperbolic node embeddings, as such embeddings frequently yield lower distortion than their Euclidean analogs. This has led to a number of graph machine learning papers that incorporate non-Euclidean representations and resulted in a number of highly influential papers, namely Ganea et al. (2018) and Chami et al. (2019), both of which later inspired even more related geometric graph machine learning work (Lou et al., 2020; Chen et al., 2021; Zhang et al., 2019; 2021b). This line of work, however, has difficulty when generalizing beyond the simplistic geometric structure of hyperbolic space, most suitable for representation of trees (Sarkar, 2011). In particular, more general graph types do not exhibit the same simplicity, making it harder to apply these methods in a principled manner.

**Graph neural network evaluation** Various pitfalls of graph neural network evaluation have been investigated and addressed, specifically those concerning the necessity of uniform train-validation-test splits and consistent early stopping criteria (Shchur et al., 2018). Further still, it has been found that more simple GNN models can frequently perform as well if not better than more complex models if they are tuned well enough (Shchur et al., 2018). Additionally, several papers have proposed neural networks that remove explicit graph usage, but still make use of graph information via graph distillation (Luo et al., 2021; Zhang et al., 2021a), showing that good performance can be obtained without explicit message passing. Despite this, our paper is the first to explicitly and rigorously investigate the necessity of graph information (as a whole) for benchmark tasks.

For many of the mathematically sophisticated techniques described above, empirical progress remains unclear. In particular, recent works have argued for greater nuance and more careful analysis in the context of these graph learning research directions. For example, Tortorella & Micheli (2023) questions the efficacy of topological rewiring for downstream tasks, Bechler-Speicher et al. (2024) shows that the inherent graph structure may not be necessary for graph classification tasks, and Katsman & Gilbert (2024) demonstrates that a well-trained simple Euclidean MLP can often beat hyperbolic models with a similar number of parameters, even on tasks previously deemed to be the most hyperbolic.

## 3    Necessity of Graph Structure in Graph Learning

Although various pitfalls of graph neural network evaluation have been addressed (Shchur et al., 2018), our paper makes a surprising discovery about canonical benchmark graph learning datasets. When we properly tune a basic MLP that uses the features alone and makes no use of graph structure, we obtain results that are comparable to those of most graph neural network methods. This is surprising because the reported performance of MLPs on graph tasks is often much weaker (e.g., see the MLP results in Liu et al. (2020),

| | Model | Amazon Computers | Amazon Photo | Coauthor CS | Coauthor Physics |
|---|---|---|---|---|---|
| Basic | LogReg (Liu et al., 2020) | $64.1_{\pm 5.7}$ | $73.0_{\pm 6.5}$ | $86.4_{\pm 0.9}$ | $86.7_{\pm 1.5}$ |
| | LabelProp (Thomas, 2009) | $70.8_{\pm 8.1}$ | $72.6_{\pm 11.1}$ | $73.6_{\pm 3.9}$ | $86.6_{\pm 2.0}$ |
| | LabelProp NL (Thomas, 2009) | $75.0_{\pm 2.9}$ | $83.9_{\pm 2.7}$ | $76.7_{\pm 1.4}$ | $86.8_{\pm 1.4}$ |
| NN | MLP (original) (Liu et al., 2020) | $44.9_{\pm 5.8}$ | $69.6_{\pm 3.8}$ | $88.3_{\pm 0.7}$ | $88.9_{\pm 1.1}$ |
| | MLP (tuned) | $82.6_{\pm 0.6}$ | $88.0_{\pm 1.0}$ | $92.8_{\pm 0.2}$ | $94.8_{\pm 0.4}$ |
| Graph NN | GCN (Kipf & Welling, 2017) | $82.6_{\pm 2.4}$ | $91.2_{\pm 1.2}$ | $91.1_{\pm 0.5}$ | $92.8_{\pm 1.0}$ |
| | GAT (Velickovic et al., 2018) | $78.0_{\pm 19.0}$ | $85.7_{\pm 20.3}$ | $90.5_{\pm 0.6}$ | $92.5_{\pm 0.9}$ |
| | MoNet (Monti et al., 2016) | $83.5_{\pm 2.2}$ | $91.2_{\pm 1.3}$ | $90.8_{\pm 0.6}$ | $92.5_{\pm 0.9}$ |
| | SAGE-mean (Hamilton et al., 2017) | $82.4_{\pm 1.8}$ | $91.4_{\pm 1.3}$ | $91.3_{\pm 2.8}$ | $93.0_{\pm 0.8}$ |
| | SAGE-maxpool (Hamilton et al., 2017) | – | $90.4_{\pm 1.3}$ | $85.0_{\pm 1.1}$ | $90.3_{\pm 1.2}$ |
| | SAGE-meanpool (Hamilton et al., 2017) | $79.9_{\pm 2.3}$ | $90.7_{\pm 1.6}$ | $89.6_{\pm 0.9}$ | $92.6_{\pm 1.0}$ |
| | DAGNN (Liu et al., 2020) | $84.5_{\pm 1.2}$ | $92.0_{\pm 0.8}$ | $92.8_{\pm 0.9}$ | $94.0_{\pm 0.6}$ |
| | CGT (Hoang & Lee, 2023) | $91.5_{\pm 0.6}$ | $\mathbf{95.8}_{\pm 0.8}$ | – | – |
| | Exphormer (Shirzad et al., 2023) | $\mathbf{91.6}_{\pm 0.3}$ | $95.3_{\pm 0.4}$ | $\mathbf{95.8}_{\pm 0.2}$ | $\mathbf{97.2}_{\pm 0.1}$ |

Table 1: Above we report the node classification test accuracy for several canonical graph datasets, namely the two standard co-purchase datasets Amazon Computers and Amazon Photo (Shchur et al., 2018) and the two co-author datasets Coauthor CS and Coauthor Physics (Shchur et al., 2018). Means and standard deviations are given over 5 trials. After thorough tuning, the MLP nearly matches or exceeds most graph methods for *all* tasks, in particular outperforming almost all graph methods on Coauthor Physics. This indicates that these graph tasks can be nearly solved without leveraging graph structure, bringing into question their efficacy for measuring the performance of novel graph neural networks.

which we give in the "MLP (original)" row in Table 1), and improving this baseline has serious implications regarding the tasks.

We report node classification test accuracy on the following four widely used graph machine learning benchmark datasets: Amazon Computers, Amazon Photo, Coauthor CS, and Coauthor Physics (Shchur et al., 2018). The results are given in Table 1. We use the versions of these datasets from the widely available Deep Graph Library (DGL) (Wang et al., 2019). Our tuned MLP results are given in the "MLP (tuned)" row, while the other benchmarks are taken from prior work, specifically Liu et al. (2020), Hoang & Lee (2023), and Shirzad et al. (2023). Means and standard deviations are reported for all results over 5 trials.

The most shocking result is an improvement from 44.9% to 82.6% for the MLP on the Amazon Computers dataset. This is an improvement of 6.5 standard deviations over the original reported results in Liu et al. (2020)! The results for Amazon Photo showcase an improvement of over 4.8 standard deviations, and on Coauthor CS and Coauthor Physics, our results show improvements of over 6.4 and 5.3 standard deviations, respectively. On all datasets, the tuned MLP becomes virtually indistinguishable from most graph neural networks, and, on Coauthor Physics, the tuned MLP outperforms almost all graph neural networks except for one.

While this generally highlights the need for properly tuning fundamental baselines, we argue that this further begs the question: *are these datasets appropriate for benchmarking graph neural networks, and can more appropriate datasets be found?* Ideally, datasets meant for graph machine learning would necessitate usage of the graph to obtain nontrivially high performance (relative to that of a properly tuned MLP). We see this is *not* the case for this collection of datasets. In service of investigating this issue and finding more appropriate benchmark datasets, we conduct the analysis given in the following several sections.

Although the results in Table 1 suggest that many popular datasets can be nearly solved with an MLP, we find that this is not always the case and several examples serve as an exception. A subset of these examples are given in Table 2, where we test on the Cora, CiteSeer, and PubMed datasets (Sen et al., 2008). Once more, means and standard deviations are given across 5 trials, the metric reported is test accuracy, and we use the versions of these datasets from the DGL (Wang et al., 2019). Although we find significant improvement on all three datasets, Cora and CiteSeer are far from trivially solved by the tuned MLP. PubMed, similar to what we saw in Table 1, is almost solved by a properly tuned MLP. That being said, for two of these three widely used citation network datasets, there is a substantial statistically significant gap between the MLP and graph methods, indicating that these are potentially reasonable benchmark datasets for measuring graph neural network performance.

| | Model | Cora | CiteSeer | PubMed |
|---|---|---|---|---|
| NN | MLP (original) (Liu et al., 2020) | $61.6_{\pm 0.6}$ | $61.0_{\pm 1.0}$ | $74.2_{\pm 0.7}$ |
| | MLP (tuned) | $69.4_{\pm 1.3}$ | $66.0_{\pm 0.5}$ | $86.8_{\pm 0.4}$ |
| Graph NN | ChebNet (Defferrard et al., 2016) | $80.5_{\pm 1.1}$ | $69.6_{\pm 1.4}$ | $78.1_{\pm 0.6}$ |
| | GCN (Kipf & Welling, 2017) | $81.3_{\pm 0.8}$ | $71.1_{\pm 0.7}$ | $78.8_{\pm 0.6}$ |
| | GAT (Velickovic et al., 2018) | $83.1_{\pm 0.4}$ | $70.8_{\pm 0.5}$ | $79.1_{\pm 0.4}$ |
| | APPNP (Klicpera et al., 2018) | $83.3_{\pm 0.5}$ | $71.8_{\pm 0.4}$ | $80.1_{\pm 0.2}$ |
| | SGC (Wu et al., 2019a) | $81.7_{\pm 0.1}$ | $71.3_{\pm 0.2}$ | $78.9_{\pm 0.1}$ |
| | DAGNN (Liu et al., 2020) | $84.4_{\pm 0.5}$ | $73.3_{\pm 0.6}$ | $80.5_{\pm 0.5}$ |
| | SSP (Izadi et al., 2020) | $\mathbf{90.2}_{\pm 0.6}$ | $80.5_{\pm 0.1}$ | $87.8_{\pm 0.2}$ |
| | ACM-Snowball-3 (Luan et al., 2021) | $89.6_{\pm 1.6}$ | $\mathbf{81.3}_{\pm 1.0}$ | $\mathbf{91.4}_{\pm 0.4}$ |

Table 2: Above we give graph task results for three standard citation network datasets. The metric reported is test accuracy and the task is node classification. Means and standard deviations are given over 5 trials. After thorough tuning, the MLP nearly solves the PubMed dataset and significantly improves for both Cora and CiteSeer, though a substantial gap remains between the tuned MLP and various graph methods for those two datasets.

## 4 Feature Study

We hypothesize that the phenomenon observed in Table 1 is as a result of the fact that the features themselves contain enough of the graph structure to reduce the benefit one obtains from a graph-based approach. To investigate this hypothesis further, we conduct a feature study of the Amazon Computers dataset, which exhibited the largest increase after MLP tuning in Table 1. Our feature study was conducted as follows. The Amazon Computers dataset has 767 features (Shchur et al., 2018), so we form seven graph datasets, each of which shares the original graph from Amazon Computers while the nodes have an increasing subset of the features (in increments of 100 features). We name these datasets $\{$Amazon Computers $- n | n \in \{100k | k \in [7]\}\}$.[2] We then thoroughly tune an MLP and GCN on these seven datasets and note the performances shown in Figure 1a, also adding the final data point over 767 features from Table 1.

There are two details immediately worthy of note: on the datasets with few features, like Amazon Computers-100, the GCN already starts very high (85%+) while the MLP starts low. As the number of features increases, the GCN improves slightly, but the MLP improves much more, narrowing the gap between the two from 30%+ to less than 6%.[3] This shows that whatever benefit the graph information was yielding the GCN initially is also enjoyed by the MLP with a higher number of features; that is, the features "leak" the graph information as there are more and more of them, and the benefit is not orthogonal to whatever benefit is derived from the graph, i.e., the GCN performance does not increase to widen the gap and rather stays nearly the same throughout.

---

[2]We use the notation $[n] = \{k \in \mathbb{N} : k \leq n\}$.

[3]Note that our GCN obtains a slightly higher result than that reported in Liu et al. (2020), likely due to better tuning.

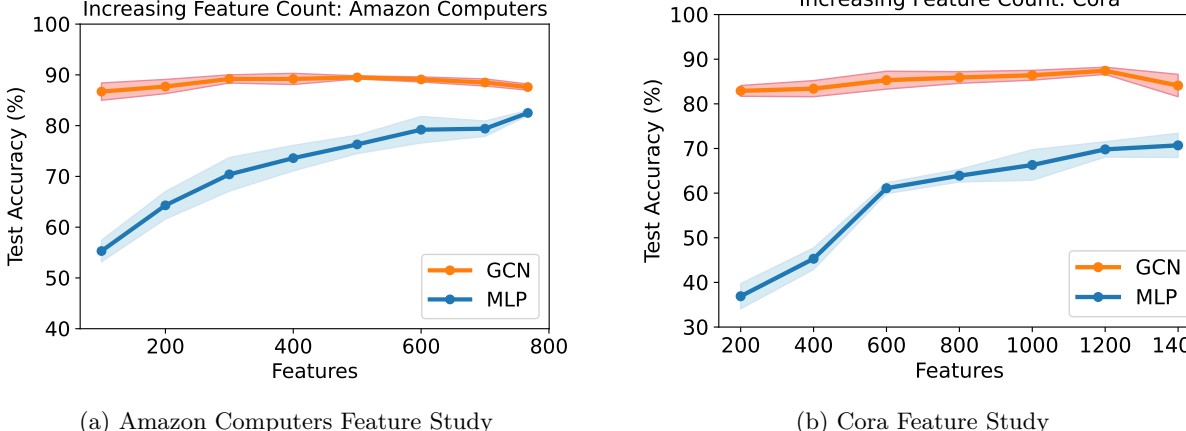

(a) Amazon Computers Feature Study  (b) Cora Feature Study

Figure 1: We conduct a feature study on the Amazon Computers and Cora datasets by synthesizing datasets with an increasing number of features. These datasets comprise the ticks along the "Features" axis. On each dataset, we thoroughly tuned the MLP and GCN. Means over 5 trials are reported, and the shaded region indicates one standard deviation. As is clearly visible, there is initially a large gap between the MLP and GCN in both subfigures, yet the gap closes considerably for Amazon Computers as the number of features increases, whereas the same is not true to the same extent for Cora. This indicates that graph information "leaks" via the features for Amazon Computers and explains why in some graph learning contexts the graph is unnecessary to obtain good performance.

As a negative control test, we perform the same experiment on the Cora dataset. The Cora dataset has 1433 features (Sen et al., 2008), so we form 7 graph datasets, each of which shares the original graph from Cora, while increasing the subset of the features (in increments of 200 features). We name these datasets $\{\text{Cora} - n | n \in \{200k | k \in [7]\}\}$. Similar to the above, we then tune an MLP and GCN on these seven datasets and note the performances shown in Figure 1b. The performance of the MLP in this instance still improves as we add more features, however the gap between the GCN and MLP remains substantial throughout. Namely, the gap remains consistently at 15-20% for feature counts past 600. This indicates that the features do not "leak" the graph information in the way they do for Amazon Computers in Figure 1a.

Our feature analysis show that there are contexts in which the features of graph datasets "leak" graph information and reduce (sometimes obviate) the need for the graph, further elucidating and explaining the results in Section 3. One must be mindful of this fact when benchmarking graph learning methods, seeking to use harder datasets that necessitate use of the graph.

## 5  Synthetic Benchmark Datasets

Given the observed phenomenon in Section 3, our goal in this section is to consider benchmark dataset design from first principles and introduce challenging synthetic graph benchmark datasets that necessitate graph information for nontrivial performance.

**Introductory definitions** To start, we define the following simple class of graph datasets. First, denote the set of all finite graphs by $\mathcal{G}$, where $G \in \mathcal{G}$ is given by $(V, E)$, where $V \subset \mathbb{N}$ is a finite set of vertices and $E$ is the associated set of edges. We define a graph dataset by a graph $G$ and associated feature vectors $\{x_i\}$, one $x_i$ for each $v_i \in V$.[4] Then, given a distribution $\mathcal{D}$ over $\mathbb{R}^d$, we define:

$$\mathcal{G}_{\mathcal{D}} = \{(G, \{x_i\}) : G \in \mathcal{G}, x_i \overset{\text{i.i.d.}}{\sim} \mathcal{D}\} \tag{1}$$

---

[4]This definition may sometimes include a set of node labels, $\{\ell_i\}$, which is something we do not consider in the context of this paper.

That is to say, $\mathcal{G}_{\mathcal{D}}$ is the set of all graph datasets whose features are drawn i.i.d. from $\mathcal{D}$. Notice that it is enough to specify a graph $G$ and a distribution $\mathcal{D}$ to obtain a graph dataset from this set.

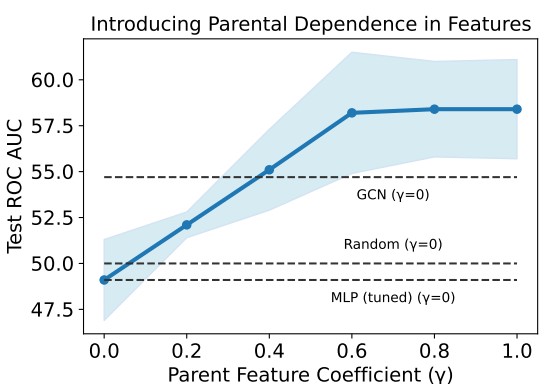

| Model | WS1000 ($\gamma = 0$) |
|---|---|
| Random | $50.0_{\pm 0.0}$ |
| MLP (tuned)[5] | $49.1_{\pm 2.2}$ |
| GCN (Kipf & Welling, 2017) | $\mathbf{54.7}_{\pm 0.4}$ |

Table 3: Link prediction results (test ROC AUC) for our synthesized WS1000 dataset, meant to illustrate a hard learning context in which a simple MLP fails, yet a graph-based GCN obtains a non-trivial performance increase.

Figure 2: Introducing parental dependence in node features very quickly leads the MLP to improve on link prediction. Each data point is obtained by tuning the MLP on the relevant synthetic $WS1000_\gamma$ dataset. The average of 5 trials is reported and the error region specifies one standard deviation.

**Synthetic Watts-Strogatz dataset** Notice that the above class of graph datasets is very simple because the features are all drawn i.i.d., completely independent of the graph structure. We can make use of a dataset with these properties to benchmark the degree to which a given graph learning method uses the graph, since the features are completely uninformative. This would produce a very challenging synthetic benchmark dataset.

To instantiate a concrete dataset, for $G$ we will select a Watts-Strogatz graph, since these graphs are a model of small world graphs (Watts & Strogatz, 1998) seen in a number of real-world contexts (e.g. citation datasets). We sample a graph $G_{WS}$ using the Watts-Strogatz model with $N = 1000$ nodes, mean degree $K = 4$, and $\beta = 0.5$. Further, we select $\mathcal{D} = \mathcal{N}(0, I_{1000})$ and sample $\{x_i\}_{i \in [1000]}$ as features, i.i.d. from $\mathcal{D}$. We give this graph dataset $(G_{WS}, \{x_i\}_{i \in [1000]})$ the name $WS1000$. Note this dataset is immediately suitable for link prediction (since this label is intrinsically derived from the graph).

We thoroughly tune the MLP and GCN on $WS1000$, giving results in Table 3. As is clearly visible, the MLP fails on this task (with a result of $49.1 \pm 2.2$ test ROC AUC) since it uses the features alone, yet the GCN obtains a nontrivial improvement of $54.7 \pm 0.4$ test ROC AUC. Although the GCN performance is not ideal, even this simple hard synthetic dataset clearly separates the MLP and the GCN. This demonstrates the capability of a graph method to exploit graph structure in a small-world context given uninformative features.

**Introducing graph structure in features** Although the above test dataset is hard and can separate the MLP from the GCN, it is, in a very direct sense, quite extreme: the features are completely uninformative and offer no discernible distinction between nodes. As a consequence, we will define the following parametric family of graph datasets that weaves graph structure into the feature design.

Let a connected graph $G = (V, E)$ be given. We define the parametric family of distributions of graph datasets $G(\mathcal{D}, v, \gamma, \nu)$ as follows, where $\mathcal{D}$ is a distribution of $\mathbb{R}^d$, $v \in V$ is a fixed vertex, and $\gamma, \nu \in \mathbb{R}$ are constants. Sampling from this family amounts to first sampling a feature $x_v \sim \mathcal{D}$ for the node $v$. Features for the remaining nodes are obtained via the following procedure, during which we traverse the graph breadth-first starting from $v$. Let $d_i$ be the shortest distance between node $v_i$ and $v$ and let $D_i$ be the set of nodes at a distance $i$ from $v$. For $i = 1, \ldots, \mathrm{diam}(G)$, we repeat the following procedure. We let $x_{v_j}$ be the feature vector corresponding to $v_j$ and $p(x_{v_j})$ be the feature vector of the "parent" node $p(v_j)$ that precedes $v_j$ during a

---
[5]Performance can be worse than random due to poor seeds that bring the average below 50 and overfitting.

| Model | Musae-Twitch | Twitch-PTBR | Musae-Facebook |
|---|---|---|---|
| MLP (tuned) | $56.5_{\pm 4.1}$ | $60.1_{\pm 4.3}$ | $50.3_{\pm 0.5}$ |
| GCN (Kipf & Welling, 2017) | $\mathbf{82.8}_{\pm 0.8}$ | $\mathbf{85.4}_{\pm 0.4}$ | $\mathbf{90.6}_{\pm 0.4}$ |

Table 4: Link prediction results (test ROC AUC given) for three additional benchmark datasets that showcase good separation between a meticulously tuned MLP and GCN, indicating that there is considerable benefit to be had from using the graph, thereby making these datasets suitable for benchmarking.

breadth-first traversal of the graph starting at $v$:

$$\forall v_j \in D_i : x_{v_j} \leftarrow \gamma p(x_{v_j}) + \nu z, z \sim \mathcal{D}$$

We see that this procedure amounts to sampling a graph structure-derived set of features for the graph. This procedure is similar albeit more general than the one given in Katsman & Gilbert (2024) since it works over unrooted graphs.

**Synthetic $WS1000_\gamma$ datasets** Using our above definitions, we generate synthetic graph datasets that share the graph WS1000. That is, the datasets will differ only in terms of their features. In particular, we will sample from the parametric set of distributions $WS(\mathcal{N}(0, I_{1000}), v_0, \gamma, 1)$ for various values of $\gamma$, where we selected $v_0 \sim \mathcal{U}(V)$ uniformly at random from the vertex set $V$ of WS1000. Note that in this context, the vertex $v_0$ is simply a fixed arbitrary "root" and $\gamma$ controls the level of "parental" dependence during the feature generation process. We sample a graph dataset for each of the values of $\gamma$ in $\{0.2k | k \in [5]\}$ and call the resulting set of 5 graph datasets $\{WS1000_\gamma | \gamma \in \{0.2k | k \in [5]\}\}$. We tune the MLP for each of these datasets and showcase the results in Figure 2. Note that as parental dependence increases in the features, the MLP begins to outperform the GCN, yet the tasks remains challenging, with the highest attained performance being approximately 60% test ROC AUC for $\gamma \geq 0.6$.

With the $WS1000_\gamma$ datasets, we have introduced a collection of hard graph datasets over a small-world Watts-Strogatz network where the features capture some of the graph structure, thereby yielding a more realistic yet still quite challenging benchmark setting.

As an aside, one can view this sampling procedure as generating a Euclidean embedding for the graph that is then used as the feature set. Although it is not always possible to embed a graph in Euclidean space well (Nickel & Kiela, 2017), this example and the results in Figure 2 show that this sampling procedure can be sufficient to obtain the equivalent benefit of a GCN and even more so.

## 6 Suggested Benchmarks

In addition to the synthetic benchmarks we give in Section 5, we highlight several real-world datasets (as measured by MLP performance) that we believe make for good graph learning benchmark tasks. In addition to Cora from Section 3, we recommend Musae-Twitch, Twitch-PTBR, and Musae-Facebook (Rozemberczki et al., 2019). On each of these datasets, we hyperparemeter tuned the MLP and GCN (as a representative graph method) to do link prediction. The results are shown in Table 4. Note that despite considerable tuning, the MLP remains much lower than the GCN in all cases. Also please note that the task we use to separate the MLP and GCN here for these datasets is link prediction, which is intrinsically derived from the graph and is generally considered to be more basic and fundamental than node classification (which is typically a more challenging task).

## 7 Discussion

**Importance of Hyperparameter Tuning in Benchmarking** Our results point to the fact that the simple feature-only MLP seems to have been systemically under-tuned (resulting in lackluster performance) for multiple papers. This lack of strong baselines is largely deleterious for these benchmarks, since earlier work could have discarded these faulty benchmarks sooner.

Furthermore, we note that since thoroughly tuning hyperparameters for all models would far exceed our compute budget, we report results from preexisting papers for all models other than the MLP in Tables 1 and 2. As such, we do not claim that other graph neural network results have been perfectly tuned. In particular, we demonstrated in Figures 1a and 1b that even the GCN benchmark can be improved (although only slightly) with careful hyperparameter tuning. This can pose an issue, as it is possible that some highly performant graph methods may largely derive their benefit from hyperparameter tuning as opposed to methodological superiority (in fact, this has been shown for a number of methods in Shchur et al. (2018)), which obfuscates the discernment of truly superior graph models on datasets where the graph actually matters.

**Do Graph Networks Really Need Graphs?** Although a central point of our paper is that graph neural networks do not seem to fare much better than feature-only methods for several common graph tasks, we emphasize that it would be incorrect to draw a general conclusion that *all* graph neural networks do not offer meaningful benefits. Indeed, our results even show that more sophisticated graph neural networks tend to outperform feature-only methods. However, we emphasize that without proper benchmarking, the extent of these improvements is unclear.

**Limitations** Our current analysis primarily focuses on seven popular graph datasets. We anticipate that similar conclusions may apply to numerous other datasets, potentially including those from the Open Graph Benchmark (Hu et al., 2020). However, we did not attempt to conduct an analysis on these datasets due to the resource-intensive nature of the tuning process on larger graphs. Nevertheless, our central conclusions remain valid based on the seven datasets we used. Additionally, we deal primarily with the intragraph prediction tasks of node classification and link prediction. We do not investigate graph machine learning for graph classification, despite the fact that there is already early evidence that similar conclusions may hold in that context: Bechler-Speicher et al. (2024) note that using an empty graph in the context of graph classification improves performance for GNNs on two graph classification datasets.

**Future work** While our work sheds light on several problems with existing graph learning benchmarks, we emphasize that much future work remains. For example, the exact relationship between graph structure and downstream performance is not yet fully understood. Coming up with a metric to properly characterize graph datasets, in particular the interaction of the features and graph as it relates to suitability of the dataset for graph learning, is an excellent avenue for future work. Our approach in this paper provides a coarse and somewhat expensive tool to measure the relationship between graph dataset structure and downstream performance; future work should be geared towards better determining this relationship without the need to sweep hyperparameters for an MLP.

**Impact Statement** This paper deals with challenges in evaluating graph learning benchmarks. While there are no direct societal implications of our work (since our constructions happen mostly at a meta-scientific level), graph learning on social networks has the potential for both positive (i.e. community building) and negative (i.e. creating echo chambers) societal ramifications. However, these are ultimately out of scope for our work.

## 8 Conclusion

In this paper we tested seven canonical graph datasets and identified five for which a well-tuned feature-only MLP drastically improves upon previously presented baseline performance. In particular, for many of these results, the well-tuned MLP is able to surpass the reported results for many graph neural networks, indicating a fundamental failure in the use of these datasets for the purpose of benchmarking graph neural networks. We further analyzed these failure cases, showing that the phenomenon we observed stems from a leakage of graph structure in the node features themselves. Finally, we presented several synthetic and real world test tasks that explicitly mitigate said graph structure contamination. We believe this collection of datasets can serve as a better set of benchmarks for measuring the performance of novel graph neural networks.

We hope that our work draws attention to fundamental challenges with graph neural networks and graph network benchmarks. Ultimately, gaining a complete understanding of the relationship between the node features and graph structure is what will pave the way for more rigorous development of graph networks in contexts for which they are appropriate.

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

# Appendix

In the supplementary material for this paper, we present feature study results on additional datasets, give additional information about the synthetic datasets we introduce in the main paper, and provide further experimental details. All code and synthesized datasets are also provided. Precise commands for reproducing results of the main paper are given in the `README.md` file provided in the supplementary material.

## A   Feature Study: Results for Additional Datasets

In this section we give feature study results for 5 additional datasets from Section 3: Amazon Photo (Shchur et al., 2018), Coauthors CS (Shchur et al., 2018), Coauthors Physics (Shchur et al., 2018), Pubmed (Sen et al., 2008), and CiteSeer (Sen et al., 2008).

The Amazon Photo dataset has 745 features (Shchur et al., 2018), so we form seven graph datasets, each of which shares the original graph from Amazon Photo while the nodes have an increasing subset of the features (in increments of 100 features). We name these datasets $\{\text{Amazon Photo} - n | n \in \{100k | k \in [7]\}\}$. We then thoroughly tune an MLP and GCN on these seven datasets and note the performances shown in Figure 3a. As we can somewhat anticipate based on the results given in Section 3, the gap between the MLP and GCN closes considerably as the number of features increases.

We continue with the Pubmed dataset; the Pubmed dataset has 500 features (Sen et al., 2008), so we form five graph datasets, each of which shares the original graph from Pubmed while the nodes have an increasing subset of the features (in increments of 100 features). We name these datasets $\{\text{Pubmed} - n | n \in \{100k | k \in [5]\}\}$. We then thoroughly tune an MLP and GCN on these five datasets and note the performances shown in Figure 3b. The gap between the MLP and GCN closes entirely as the number of features increases, indicating that the features essentially subsume additional information from the graph.

We perform a similar analysis with the Coauthor Physics dataset; this dataset has 8415 features (Shchur et al., 2018), so we form eight datasets, each of which shares the original graph from Coauthor Physics while the nodes have an increasing subset of the features (in increments of 1000 features). We name these datasets $\{\text{Coauthor Physics} - n | n \in \{1000k | k \in [8]\}\}$. We then thoroughly tune an MLP and GCN on these eight datasets and note the performances shown in Figure 3c. As we can somewhat anticipate based on the results given in Section 3, the gap between the MLP and GCN closes considerably as the number of features increases, with the MLP matching the performance of the GCN for $\geq 7000$ features.

We perform a feature study on the Coauthor CS dataset; this dataset has 6805 features (Shchur et al., 2018), so we form six datasets, each of which shares the original graph from Coauthor Physics with the nodes have an increasing subset of the features (in increments of 1000 features). We name these datasets $\{\text{Coauthor CS} - n | n \in \{1000k | k \in [6]\}\}$. We then thoroughly tune an MLP and GCN on these six datasets and note the performances shown in Figure 3d. The gap between the MLP and GCN closes considerably as the number of features increases, and in fact, somewhat surprisingly, the MLP exceeds the performance of the GCN for $\geq 5000$ features. This indicates that the graph structure may actually be somewhat harmful for these large numbers of features, which is somewhat similar to an observation made in the context of graph classification by Bechler-Speicher et al. (2024) (i.e. that the graph structure can sometimes be harmful).

Lastly, we perform a feature study on the CiteSeer dataset; this dataset has 3703 features (Sen et al., 2008), so we form six datasets, each of which shares the original graph from CiteSeer while the nodes have an increasing subset of the features (in increments of 600 features). We name these datasets $\{\text{CiteSeer} - n | n \in \{600k | k \in [6]\}\}$. We then thoroughly tune an MLP and GCN on these six datasets and note the performances shown in Figure 3e. The gap between the MLP and GCN closes considerably as the number of features increases. This indicates that the features essentially subsume some of the graph information.

## B   Synthetic Dataset Information

As was mentioned in Section 5, we are releasing 6 synthetic datasets as a part of this paper. Following the notation of that section, these are $WS1000$, together with the five datasets that have variable $\gamma$: $WS1000_{0.2}$,

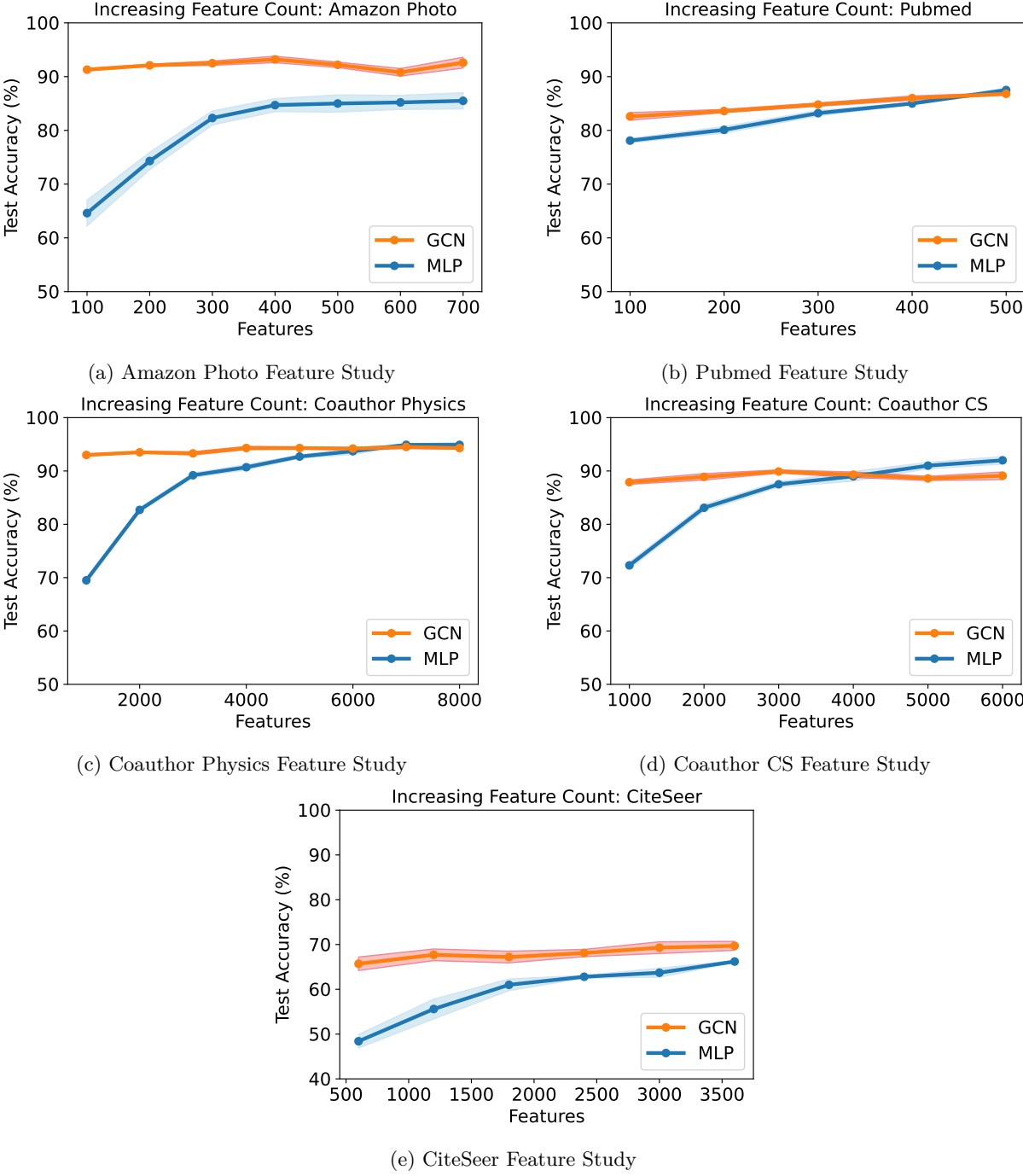

(a) Amazon Photo Feature Study

(b) Pubmed Feature Study

(c) Coauthor Physics Feature Study

(d) Coauthor CS Feature Study

(e) CiteSeer Feature Study

Figure 3: We conduct feature studies on the Amazon Photo, Pubmed, Coauthor Physics, Coauthor CS, and CiteSeer datasets by synthesizing datasets with an increasing number of features. These datasets comprise the ticks along the "Features" axis. On each dataset, we thoroughly tuned the MLP and GCN. Means over 5 trials are reported, and the shaded region indicates one standard deviation. As is clearly visible, there is a gap between the MLP and GCN in all subfigures, yet the gap closes considerably as the number of features increases. This indicates that graph information "leaks" via the features and explains why in some graph learning contexts the graph is unnecessary to obtain good performance. Somewhat surprisingly, we even see the MLP match and outperform the GCN in some cases involving Coauthor Physics and Coauthor CS.

$WS1000_{0.4}$, $WS1000_{0.6}$, $WS1000_{0.8}$, $WS1000_{1.0}$. All of these datasets share the same 1000-node graph, sampled by way of the Watts-Strogatz model (Watts & Strogatz, 1998) and differ in their node features as specified in Section 5. Node features are 1000-dimensional real-valued vectors associated with every node of the graph. Our datasets are intended to be used for link prediction, as hard benchmark tasks for graph machine learning methods.

All six of our datasets are released under a CC0 1.0 Universal License; download instructions are provided in the supplementary material under the `syntheticdatasets` folder. The code necessary for synthesis is provided as well. The graph is provided via a simple edges CSV file and the features are provided using the NPZ compression format, and are easily loaded via the `np.load()` function (Harris et al., 2020). After the reviewing period, we plan to release our code and datasets via Github hosting so they will be available to the public indefinitely.

## C  Experimental Details

In this section, we discuss concrete experimental details of our paper. All experiments were run with the help of two RTX 3090 GPUs and we estimate a total of 4000 GPU hours went into obtaining the results given in this paper. Extensive hyperparameter tuning for the MLP was conducted by performing Bayesian sweeps with the help of the Weights & Biases platform. An example sweep file with relevant hyperparameters is given in the attached supplementary material at `graphdatasets_hgcn/example-sweep-lp-watts1000-k4-p05-g00-bayes-mlp.yml`. The MLP hyperparameters found in this file were unchanged for all MLP sweeps conducted. Exact commands for reproducing the results found in our main paper using our provided code can be found in the supplementary material via the `README.md` file.

