# OpenReview forum: "Revisiting the Necessity of Graph Learning and Common Graph Benchmarks"
_TMLR — Rejected by TMLR_

### Review · Reviewer_TwXE · 2025-03-05

**Summary Of Contributions:**

The authors demonstrate that, for some widely used datasets, the performance of MLP is comparable to or even better than that of graph based methods. Specifically, the authors test seven commonly used datasets and a well-tuned MLP performs better than at least one GNN in five of which. This suggests that the graph structure may not be as crucial as previously thought. The authers perform an analysis to understand why node features alone can suffice for these tasks. Besides, the authors propose some approaches to benchmark the effect of graph layers.

**Audience:**

Yes

**Broader Impact Concerns:**

The impact statement in this article has fully considered the potential social and ethical implications of the research, therefore there are no further concerns.

**Claims And Evidence:**

Yes

**Requested Changes:**

1.  The authors should consider expanding their analysis to include more datasets, to ensure that their conclusions are more generalizable.
2.  The authors should provide a simple theoretical analysis on why certain datasets benefit more from graph structure than others.
3. While the focus on node classification and link prediction is justified, the authors could briefly discuss how their findings might apply to other graph learning tasks.

**Strengths And Weaknesses:**

Strengths:
The paper presents a thorough empirical analysis. The findings are interesting. The feature study provides valuable insights into why node features alone can perform well on certain datasets, offering a deeper understanding of the interplay between features and graph structure.

Weaknesses:
1. While the paper analyzes seven popular datasets, the conclusions might not generalize to all graph learning tasks, especially those involving larger or more complex graphs.
2. The paper primarily focuses on node classification and link prediction tasks. It does not address other important graph learning tasks such as graph classification or graph generation, where the necessity of graph structure might differ.
3. While the empirical results are strong, the paper lacks a theoretical framework to explain why certain datasets benefit more from graph structure than others. A deeper theoretical analysis could strengthen the paper's contributions.

---

> ### Author Response · Authors · 2025-04-08
> **Author Response (1 of 2)**
>
> Thank you for the review and the constructive comments! We appreciate that you think our paper “presents a thorough empirical analysis” and that our “findings are interesting.” We address your comments from the “Weaknesses” and “Requested Changes” sections below.
>
> ---
>
> > While the paper analyzes seven popular datasets, the conclusions might not generalize to all graph learning tasks, especially those involving larger or more complex graphs.
>
> A: This is a good point, however, our paper crucially focuses on these datasets as they are widely used and cited benchmarks. If we look at the paper that introduced the four datasets in Table 1 of our paper, it has been cited 1600+ times with 600+ of those citations appearing since 2024 (see Google scholar for this: https://scholar.google.com/scholar?as_ylo=2024&hl=en&as_sdt=8005&sciodt=0,7&cites=17552796071161142589&scipsc=). If we look at the paper that introduced the three datasets in Table 2 of our paper, it has been cited 4900+ times with 1200+ of those citations appearing since 2024 (see Google scholar for this: https://scholar.google.com/scholar?as_ylo=2024&hl=en&as_sdt=8005&sciodt=0,7&cites=6470587456114814344&scipsc=). Given their prevalence, we decided to focus on these datasets. Moreover, the fact that we have clearly demonstrated the existence of benchmark datasets on which well-tuned MLPs outperform a number of GNNs is highly concerning for the field. We emphasize that transitioning away from these benchmarks and using more suitable ones is crucial. This, in essence, is the critical point of our paper. It is not that our conclusions hold for all graph datasets, and in fact we see that CoRA (in Table 2) benefits from graph learning as well as the datasets we suggest in Table 4.
>
> ---
>
> > The paper primarily focuses on node classification and link prediction tasks. It does not address other important graph learning tasks such as graph classification or graph generation, where the necessity of graph structure might differ.
>
> A: You are correct that we focus only on two common types of graph tasks: node classification and link prediction. That being said, node classification has long been used as a canonical graph benchmark task, dating back to the original Kipf & Welling paper that introduced the GCN. Given the widespread use of this task and related datasets for benchmarking, we decided that this was a good focus for our paper. We believe an investigation of other tasks is a great avenue for future work.
>
> ---
>
> > While the empirical results are strong, the paper lacks a theoretical framework to explain why certain datasets benefit more from graph structure than others. A deeper theoretical analysis could strengthen the paper's contributions.
>
> A: This is a good point and we believe that developing a good understanding of graph structure, features structure, and their interaction in graph datasets is very important for the future of graph learning. Our paper serves as a crucial starting point where we question the fundamental assumption that graph structure is always helpful, show that it is frequently not helpful on widely used benchmark datasets, and introduce a variety of alternative datasets that are better suited as benchmarks (as well as hard synthetic datasets, suitable as “limiting” cases). We believe that obtaining a good theoretical understanding of the interaction of graph structure and graph features is a great avenue for future work.
>
> ---
>
> > The authors should consider expanding their analysis to include more datasets, to ensure that their conclusions are more generalizable.
>
> A: Thank you for this suggestion; we focus our efforts on 7 widely cited graph benchmark datasets due to the high impact of these datasets on the field and the fact that we have the resources to perform extensive tuning on datasets of this size. We anticipate that similar conclusions may apply to numerous other datasets, potentially including those from the Open Graph Benchmark (Hu et al., 2020). We did not attempt to conduct an analysis on these datasets due to the resource-intensive nature of the tuning process on larger graphs. Nevertheless, the seven datasets we used are sufficient to show our central conclusions hold.

---

> > ### Author Response · Authors · 2025-04-08
> > **Author Response (2 of 2)**
> >
> > > The authors should provide a simple theoretical analysis on why certain datasets benefit more from graph structure than others.
> >
> > A: We believe theoretical analysis is an important part of the future work that should happen in this field and make some first steps towards this with our empirical analysis in Section 4: we show that for datasets that benefit from graph structure, such as CoRA, as we increase the number of features, the gap between MLP and GCN performance remains large, indicating that there is orthogonal benefit to be had from incorporating the features. The same does not hold for Amazon Computers, indicating that the features subsume at least some of the graph information, reducing its importance.
> >
> > ---
> >
> > > While the focus on node classification and link prediction is justified, the authors could briefly discuss how their findings might apply to other graph learning tasks.
> >
> > A: Certainly, we are honest about the fact that we deal primarily with the intragraph prediction tasks of node classification and link prediction. We do not investigate graph machine learning for graph classification, despite the fact that there is already early evidence that similar conclusions may hold in that context: Bechler-Speicher et al. (2024) note that using an empty graph in the context of graph classification improves performance for GNNs on two graph classification datasets. Analysis on other graph learning tasks is a great avenue for future work.

---

### Review · Reviewer_dLSR · 2025-03-25

**Summary Of Contributions:**

This paper investigates whether graph learning is really necessary for existing graph benchmarks.
The authors conducted experiments and discovered that a well-tuned MLP performs almost as good as many existing graph SOTAs on node and edge classification datasets.
The paper then proposed a new synthetic dataset which requires graph structural information to achieve non-trivial performance.

**Audience:**

Yes

**Broader Impact Concerns:**

The paper properly addressed its broader impact in the conclusion section in my opinion.

**Claims And Evidence:**

Yes

**Requested Changes:**

1. I suggest that the authors further discuss the experimental results in [2] to explain how [2] complements the graph classification aspect of the paper’s claim.

2. Please address Weaknesses (1) and (2) so that I can provide more detailed suggestions.

Overall, I believe the paper has strong potential for acceptance to TMLR, but I need some clarification before I can fully endorse it.


[2] Graph neural networks use graphs when they shouldn't (ICML 2024)

**Strengths And Weaknesses:**

## Strengths
1. The paper is very well-written and easy to follow.

2. The motivation is sound, reasonable, and novel.

3. The feature analysis section is informative and sufficient to support the authors’ claim (in my opinion).

4. The experiments are thoroughly conducted.

## Weaknesses

1. I believe the major weakness of this paper is whether we truly need a new synthetic dataset for graph learning. If most existing graph-based datasets do not require graph information to perform well, doesn't that raise the question of whether we need graph learning at all? Moreover, I believe other tasks involving graph generation (e.g., protein synthesis) already require graph structural information to achieve non-trivial performance, which again raises the question of whether a new dataset is necessary.

2. While the paper questions the necessity of using graph information in common graph benchmarks, it does not thoroughly investigate all types of graph-based tasks (e.g., graph classification, graph generation, community detection, etc.). The paper focuses on node and edge classification, which, in my opinion, are the least likely to require structural graph information, as information from neighboring nodes may often be sufficient.

3. Some related works could be further discussed or at least mentioned. For example, the authors should include [1]. While it may not be published in a top-tier conference, the analysis presented in that work is similar to what the authors have done. Additionally, I suggest the authors discuss the results from [2] further. [2] addresses some of the gaps in the current manuscript, particularly with regard to graph classification tasks. To thoroughly examine whether graph information is needed in common graph benchmarks, the authors should explain in detail how and why [2] complements the manuscript.


[1] Taxonomy of Benchmarks in Graph Representation Learning (LoG 2022)
[2] Graph neural networks use graphs when they shouldn't (ICML 2024)

---

> ### Author Response · Authors · 2025-04-08
> **Author Response**
>
> Thank you for the review and the constructive comments. We appreciate that you think our work has a “sound, reasonable” motivation and that “the experiments are thoroughly conducted.” We address your comments from the “Weaknesses” and “Requested Changes” sections below.
>
> ---
>
> > I believe the major weakness of this paper is whether we truly need a new synthetic dataset for graph learning. If most existing graph-based datasets do not require graph information to perform well, doesn't that raise the question of whether we need graph learning at all? Moreover, I believe other tasks involving graph generation (e.g., protein synthesis) already require graph structural information to achieve non-trivial performance, which again raises the question of whether a new dataset is necessary.
>
> A: Although it is tempting to ask whether we need graph learning at all, we emphasize that even in our paper, there are clearly cases where one benefits from use of the graph. For example, the CoRA dataset (in Table 2) clearly benefits from the graph as well as the datasets in Table 4 for link prediction. The main purpose of introducing hard synthetic datasets is to give a “limiting case” where the features are hardly informative of the graph structure, decoupling the features from the graph and forcing the algorithm to make use of the graph in order to make nontrivial progress with the task at hand. We believe that in addition to real datasets where graph structure helps, it is important to have a very hard “limiting case” that makes it apparent how much a machine learning algorithm is actually using the graph information.
>
> ---
>
> > While the paper questions the necessity of using graph information in common graph benchmarks, it does not thoroughly investigate all types of graph-based tasks (e.g., graph classification, graph generation, community detection, etc.). The paper focuses on node and edge classification, which, in my opinion, are the least likely to require structural graph information, as information from neighboring nodes may often be sufficient.
>
> A: You are correct that we focus only on two common types of graph tasks: node classification and link prediction. That being said, node classification has long been used as a canonical graph benchmark task, dating back to the original Kipf & Welling paper that introduced the GCN. Given the widespread use of this task and related datasets for benchmarking, we decided that this was a good focus for our paper. We believe an investigation of other tasks is a great avenue for future work.
>
> ---
>
> > Some related works could be further discussed or at least mentioned. For example, the authors should include [1]. While it may not be published in a top-tier conference, the analysis presented in that work is similar to what the authors have done. Additionally, I suggest the authors discuss the results from [2] further. [2] addresses some of the gaps in the current manuscript, particularly with regard to graph classification tasks. To thoroughly examine whether graph information is needed in common graph benchmarks, the authors should explain in detail how and why [2] complements the manuscript.
>
> A: Thank you for bringing the paper [1] to our attention; we will cite it and explain in the related work section how it relates to our paper. We have cited [2] and mentioned it briefly, but will expand in the related work section how it essentially makes very similar observations in the context of graph classification.
>
> ---
>
> > I suggest that the authors further discuss the experimental results in [2] to explain how [2] complements the graph classification aspect of the paper’s claim.
>
> A: Thank you for this suggestion; we will be sure to do so, as mentioned above.
>
> ---
>
> > Please address Weaknesses (1) and (2) so that I can provide more detailed suggestions.
>
> A: We have done our best to address these weaknesses above; please let us know if there are any additional concerns.

---

### Review · Reviewer_MD1v · 2025-03-28

**Summary Of Contributions:**

This work relies on empirical evidence that has been shown across many diverse graph benchmarks and papers over the last five years that for many popular graph benchmarks, MLPs can achieve the best performance and not GNNs.
This paper shows that this statement is true on four more graph benchmarks.

**Audience:**

No

**Claims And Evidence:**

No

**Requested Changes:**

Consider providing a different contribution that will provide new insights to the community.

**Strengths And Weaknesses:**

The weakness of this work is a lack of contribution. As the authors themselves review and widely cite in the intro and related work section, many works have already shown that MLPs can outperform GNNs on many graph benchmarks. Therefore, this fact is already widely known in the community, and IMHO does not require further support.
To the best of my understanding, the claimed contribution of this paper is that they show this is also true for four more datasets, and not true for three other datasets. IMHO, this cannot be considered a contribution.
First, as I explained, the fact that graph benchmarks do not necessarily need GNNs to perform best is already widely known and accepted by the community; therefore, this paper does not provide any new findings on that.
Second, the authors claim in the abstract that the 7 datasets they evaluated are "most commonly used." This statement requires some evidence.
Third, the reported datasets are not the most commonly used but rather old and fading away as benchmarks from the graph learning literature.

---

> ### Author Response · Authors · 2025-04-08
> **Author Response**
>
> We address your concerns point-by-point below.
>
> ---
>
> > The weakness of this work is a lack of contribution. As the authors themselves review and widely cite in the intro and related work section, many works have already shown that MLPs can outperform GNNs on many graph benchmarks. Therefore, this fact is already widely known in the community, and IMHO does not require further support. To the best of my understanding, the claimed contribution of this paper is that they show this is also true for four more datasets, and not true for three other datasets. IMHO, this cannot be considered a contribution.
>
> A: We are unsure what works specifically you are referring to when you say that “many works have already shown that MLPs can outperform GNNs on many graph benchmarks”. We would greatly appreciate a more precise citation so we can more accurately address this claim.
>
> The fact that this is “widely known” is a very bold claim and to the best of our knowledge false, given by the fact that researchers continue to use the datasets in our study to benchmark their models (please see the comment on citations below).
>
> To the best of our knowledge, our analysis is carefully performed, interesting, and novel. However, we would also like to note that TMLR guidelines (https://jmlr.org/tmlr/acceptance-criteria.html) explicitly emphasize that a paper should not be rejected if "a method [is] considered not 'novel enough,' as novelty of the studied method is not a necessary criteria for acceptance."
>
> ---
>
> > Second, the authors claim in the abstract that the 7 datasets they evaluated are "most commonly used." This statement requires some evidence. Third, the reported datasets are not the most commonly used but rather old and fading away as benchmarks from the graph learning literature.
>
> A: These 7 datasets are older than the newer OGB benchmarks (that we cite as Hu et al. (2020)), but were and still are widely used by the graph machine learning community at large. If we look at the paper that introduced the four datasets in Table 1 of our paper, it has been cited 1600+ times with 600+ of those citations appearing since 2024 (see Google scholar: https://scholar.google.com/scholar?as_ylo=2024&hl=en&as_sdt=8005&sciodt=0,7&cites=17552796071161142589&scipsc=). If we look at the paper that introduced the three datasets in Table 2 of our paper, it has been cited 4900+ times with 1200+ of those citations appearing since 2024 (see Google scholar: https://scholar.google.com/scholar?as_ylo=2024&hl=en&as_sdt=8005&sciodt=0,7&cites=6470587456114814344&scipsc=). Moreover, if you look at any of these datasets on Papers with Code (paperswithcode.com), you will see that many papers continue to use them as benchmarks in 2024 and 2025 (e.g. Amazon Computers https://paperswithcode.com/sota/node-classification-on-amazon-computers-1).

---

### Author Response · Authors · 2025-04-08
**Address to Reviewers**

Thank you to all the reviewers for their time and their constructive comments! Below each review, we have posted a rebuttal that directly addresses concerns. If you wish to obtain further clarification, please reply in the relevant thread, and we will get back to you as soon as possible.

---

### Decision · Action_Editor_67r8 · 2025-07-06

**Recommendation:** Reject

**Additional Comments:**

The reviewers' concerns are mainly on (i) the phenomenon "MLPs can outperform GNNs on graph benchmarks" has been studied by existing works, and (ii) the insights seem to rely on specific tasks and datasets and are not general enough. Especially regarding point (i), the current version of the submission does not sufficiently discuss the relavent findings in existing works. As a result, the claims are misleading because these insights are not as novel as the submission portrays.

I feel that the authors could consider rebalancing the paper’s content by placing greater emphasis on the feature study part of the paper and offering more comprehensive analyses on it. It is also pretty important to consider more diverse model architectures, and contemporary datasets including those from OGB and RelBench, as suggested by a reviewer.

**Audience:**

Yes

**Audience Explanation:**

This paper investigates the phenomenon that MLPs can outperform GNNs on graph benchmarks, which is an interesting topic for TMLR's audience.

**Claims And Evidence:**

No

**Claims Explanation:**

While the submission provides interesting discussions and analyses on the fact that MLPs can outperform GNNs on various graph benchmarks, two of the three reviewers still have concerns about this paper. In particular, one reviewer pointed out that the phenomenon of MLPs outperforming GNNs has already been studied in a series of existing works [1-4]. I agree with the reviewer and believe that [3,4] are especially relevant to this submission. Given this, the authors’ experiments on the seven datasets, although interesting, may offer relatively limited additional insights. Another reviewer also commented that this part of the paper is largely limited to direct observations on specific tasks and datasets. Since similar phenomena have already been explored in prior work, it may be more important for the authors to highlight their in-depth analyses, unified insights, and conclusions drawn from these experiments.

Currently, I feel that the paper does not adequately acknowledge the findings of existing works. In addition, approximately half of the main paper is devoted to presenting experimental results and comparisons across the seven datasets. The authors could consider rebalancing the paper’s content by placing greater emphasis on the feature study section and providing more comprehensive analyses in this direction. It is also important to consider a wider range of model architectures and to include contemporary datasets, such as those from OGB and RelBench, as suggested by one of the reviewers.

[1] Learning MLPs on Graphs: A Unified View of Effectiveness, Robustness, and Efficiency, ICLR 2023

[2] Pure Transformers are Powerful Graph Learners, NeurIPS 2022

[3] Graph Neural Networks Use Graphs When They Shouldn't, ICML 2024

[4] A Fair Comparison of Graph Neural Networks for Graph Classification, ICLR 2020

**Resubmission Of Major Revision:**

The authors may consider submitting a major revision at a later time.